# The Molecular Basis of FIX Deficiency in Hemophilia B

**DOI:** 10.3390/ijms23052762

**Published:** 2022-03-02

**Authors:** Guomin Shen, Meng Gao, Qing Cao, Weikai Li

**Affiliations:** 1Henan International Joint Laboratory of Thrombosis and Hemostasis, Henan University of Science and Technology, Luoyang 471023, China; mengmengg1988@126.com (M.G.); cqjt516449290@126.com (Q.C.); 2School of Basic Medical Science, Henan University of Science and Technology, Luoyang 471023, China; 3Department of Biochemistry and Molecular Biophysics, Washington University in St. Louis School of Medicine, St. Louis, MO 63110, USA

**Keywords:** hemophilia B, coagulation factor IX, molecular mechanism, γ-carboxylation, vitamin K-dependent proteins, point mutation, missense mutation, aberrant splicing, vitamin K, coagulation factor VIII

## Abstract

Coagulation factor IX (FIX) is a vitamin K dependent protein and its deficiency causes hemophilia B, an X-linked recessive bleeding disorder. More than 1000 mutations in the *F9* gene have been identified in hemophilia B patients. Here, we systematically summarize the structural and functional characteristics of FIX and the pathogenic mechanisms of the mutations that have been identified to date. The mechanisms of FIX deficiency are diverse in these mutations. Deletions, insertions, duplications, and indels generally lead to severe hemophilia B. Those in the exon regions generate either frame shift or inframe mutations, and those in the introns usually cause aberrant splicing. Regarding point mutations, the bleeding phenotypes vary from severe to mild in hemophilia B patients. Generally speaking, point mutations in the *F9* promoter region result in hemophilia B Leyden, and those in the introns cause aberrant splicing. Point mutations in the coding sequence can be missense, nonsense, or silent mutations. Nonsense mutations generate truncated FIX that usually loses function, causing severe hemophilia B. Silent mutations may lead to aberrant splicing or affect FIX translation. The mechanisms of missense mutation, however, have not been fully understood. They lead to FIX deficiency, often by affecting FIX’s translation, protein folding, protein stability, posttranslational modifications, activation to FIXa, or the ability to form functional Xase complex. Understanding the molecular mechanisms of FIX deficiency will provide significant insight for patient diagnosis and treatment.

## 1. Introduction

Hemophilia is an X-linked recessive bleeding disorder caused by a deficiency of coagulation factor VIII (FVIII) or factor IX (FIX), which are named hemophilia A (OMIM#306700) and hemophilia B (OMIM#306900), respectively. Because the genes of FVIII and FIX are located in chromosome X, hemophilia has historically been considered as a “male disease” [1]. The first hemophilia B patient, Stephen Christmas, was reported in 1952, hence hemophilia B is also called Christmas disease [2]. Compared with hemophilia A, hemophilia B is much less common and is estimated to account for 15–20% of all hemophilia cases [3]. In addition, hemophilia B is often considered less severe than hemophilia A. The prevalence of hemophilia B is about 1 in 30,000 male live births [4]. The disease severity of hemophilia B correlates with patients’ plasma level of FIX:C. Based on the FIX:C level, hemophilia B has been classified into different severities [5]. The normal level of FIX:C is 1 IU/mL in the blood plasma. Patients with <1% of normal FIX:C (<0.01 IU/mL) are defined as severe hemophilia cases and frequent occurrences of spontaneous bleeding are observed [6]. Moderate cases are patients with 1–5% (0.01–0.05 IU/mL) of normal FIX:C, and they experience bleeding primarily after injury [6]. Those with 5–40% normal FIX:C (0.05–0.40 IU/mL) are considered as mild cases, and they usually experience bleeding only with surgery or major trauma [6].

The clinical phenotype of hemophilia B, however, does not always follow these severity classifications measured from FIX:C [7]. In addition, it remains unclear why patients carrying the same mutation in some cases exhibit different bleeding tendencies that vary from mild to severe [8]. Unveiling the molecular mechanism of hemophilia B will help to understand the heterogeneity of the phenotype and provide significant insights for patient diagnosis and treatment. The enormous number of mutations identified in hemophilia B suggests that its molecular basis is extremely diverse [8,9]. In this review, we will focus on the molecular mechanism of hemophilia B. To facilitate reference to the sequence, Human Genome Variation Society (HGVS) nomenclature is applied to describe the variants of FIX—the numbering of nucleotides and proteins starts from the first methionine of FIX, with c. denoting cDNA numbering, and p. denoting protein numbering [9].

## 2. The Gene of FIX

The gene of FIX, named *F9*, was cloned in 1982 [10,11]. With the use of molecular genetic linkage analysis, *F9* was identified as the pathogenic gene of hemophilia B. The FIX gene spans 34 kb on the long arm of X chromosome at band Xq27.1 and consists of eight exons and seven introns [12,13] (Figure 1). The eight exons transcribe into a 2.8 kb mRNA (NM_000133) with a 1.4 kb noncoding region at the 3′ end [14]. The open reading frame (ORF) of the *F9* mRNA encodes a precursor protein containing 461 amino acids (aa), which include a N-terminal signal peptide (1–28 aa) and an 18-residue propeptide (29–46 aa), followed by a 415-residue mature protein that is further divided into a Gla domain (47–92 aa), two epidermal growth factor (EGF) like domains (93–171 aa), a linker sequence (172–191 aa), an activation peptide (AP, 192–226 aa), and a serine protease (SP) domain (227–461 aa) [8] (Figure 1). Exon 1 encodes a typical signal peptide designated for protein secretion from hepatocytes, exon 2 encodes the propeptide sequence and the Gla domain, exon 3 encodes the final part of the Gla domain and a short hydrophobic helical stack, exons 4 and 5 encode the two EGF-like domains, exon 6 encodes the activation peptide, and exons 7 and 8 encode the catalytic SP domain [15].

## 3. FIX Protein

FIX belongs to a group of vitamin K-dependent glycoproteins, which are synthesized in the liver. Before being secreted into the blood, it undergoes multiple intracellular processing, including the cleavage and removal of the signal peptide and propeptide sequence, γ-carboxylation of multiple glutamic acid residues in the Gla domain, partial β-hydroxylation, N-linked glycosylation, O-linked glycosylation, sulfation, and phosphorylation [16,17] (Figure 2). The post-translational modifications (PTMs) of FIX are diverse and heterogeneous, and the precise functions of most PTMs are not completely clear. Given the diverse ways that *F9* mutations lead to hemophilia B, understanding the structure-function of FIX subdomains will provide significant insight for this hereditary disease.

### 3.1. Signal Peptide and Propeptide

The signal peptide and propeptide are the regulatory sequences involved in FIX protein secretion and γ-carboxylation, respectively, and they are removed in the mature protein [18,19] (Figure 2). The signal peptide contains a hydrophobic region and a signal peptidase recognition site [8]. The hydrophobic region can be recognized by SRP, which stops FIX translation in cytosol. SRP and SRP-receptor translocate the nascent polypeptide of FIX from cytosol to the endoplasmic reticulum (ER) membrane (Figure 2, ②), and the ribosome and translocon mediate the co-translational translocation of the polypeptide into the ER lumen of the liver cells (Figure 2, ③). Subsequently, signal peptidase cleaves the signal peptide at the Cys28 position [8] (Figure 2, ④).

The propeptide is required for the γ-carboxylation of FIX, which is essential for its membrane-associated activity [18,20]. The propeptide provides the primary binding site between FIX and GGCX, which converts 12 glutamate residues in the FIX’s Gla domain to γ-carboxyglutamates (Gla) (Figure 2, ⑤) by using vitamin K hydroquinone, CO_2_, and O_2_ as cofactors [18]. After γ-carboxylation, the propeptide is cleaved off (between Arg46 and Tyr47) by protease PACE/Furin to generate the mature FIX protein in the Golgi apparatus [21,22,23] (Figure 2, ⑦).

### 3.2. Gla Domain

The Gla domain is found in vitamin K-dependent coagulation factors [20]. The γ-carboxylated Gla domain is essential for the activity of FIX and other vitamin K-dependent coagulation factors [20,24]. The binding of FIX to membranes during blood coagulation is mediated by the N-terminal Gla domain in a calcium-dependent interaction manner [25]. There are 8-9 Ca^2+^ ions bound to the Gla domain of FIX. In the absence of Ca^2+^, the structure of the Gla domain is largely disordered owing to the electrostatic repulsion in this highly negatively charged domain [26]. The binding of the Ca^2+^ ions induces a structural transition of the Gla domain, from largely unfolded and nonfunctional to tightly folded, and it is capable of membrane binding [24,25,27,28]. In the Ca^2+^-bound state, the Gla domain forms an exposed hydrophobic patch that consists of an N-terminal ω-loop and three short α-helices [25] (Figure 3A). Five Ca^2+^ ions form a robust array between the ω-loop and the first two helixes, which are further stabilized by a conserved disulfide bond, Cys64–Cys69 [25] (Figure 3A). The ω-loop binds the phospholipid-containing membrane in the Ca^2+^-bound state [24] (Figure 3A). A conformation-specific antibody, which recognizes the ω-loop region in the calcium-stabilized Gla domain of FIX, inhibits the function of FIX by interfering with its membrane binding [29,30,31]. Besides the membrane-anchoring function, the Gla domain of FIX may be involved in its binding to tissue factor (TF) in the FIX/TF/FVIIa ternary complex [32], which converts mature FIX to active FIX (FIXa) by cleaving the activation peptide off (Figure 1). Additionally, the Gla domain in FIXa contributes to its binding with the C2 domain of FVIIIa [33,34] and the collagen IV [6,35].

### 3.3. EGF-like Domains

FIX contains two EGF-like domains, EGF1 (93–129aa) and EGF2 (130–171aa), that belong to the EGF repeat superfamily. Each EGF-like domain possesses six conserved cysteine residues, which form three disulfide bonds in the order of Cys1–Cys3, Cys2–Cys4, and Cys5–Cys6 that stabilize this small domain [37,41]. The EGF-like domain is presumed to be involved in mediating protein–protein interactions [37]. Of the four loops within this domain, the third loop region may change conformation to form a canonical two-stranded anti-parallel β-sheet, serving as a modular interaction site [26]. EGF1 domain in FIX also contains a conserved Ca^2+^-binding consensus sequence [37] (Figure 3A).

The importance of EGF1 is illustrated by the large number of hemophilic mutations [41]. EGF1 participates in the activation of FIX by interacting with FXIa [42] or the TF/FVIIa complex [43]. In the FIX/TF/FVIIa ternary complex, the EGF1 domain is involved in binding to TF [43] (Figure 3A). Additionally, the EGF1 domain contributes to the FIXa interaction with FVIIIa cofactor, or functions as a spacer between the Gla domain and EGF2 that separates the phospholipid membrane, which the Gla domain interacts with, from the EGF2 and the following SP domains, by a certain distance to allow for the interaction of FIXa with FVIIIa and FX [34,44].

The EGF2 domain of FIXa, but not FIX, may be involved in binding to the platelet phospholipid membrane surface, as well as to FVIIIa and FX [44]. By using EGF2 replacement chimeras, researchers demonstrated that residues 134–155 of the EGF2 domain in FIXa mediated its binding to activated platelets and the assembly of FIXa/FVIIIa complex (also called the Xase complex) [45,46]. Further study supported that residues Asn135, Ile136, and Val153 in the EGF2 domain were essential for normal interactions of FIXa with the activated platelet surface and the assembly of the Xase complex on activated platelets [47]. The EGF2 domain in FIXa may play a minor role in binding to FVIIIa. Nevertheless, the EGF2 domain makes extensive contact with the SP domain of FIXa (Figure 3A), which may affect the interaction of the SP domain with FVIIIa by interfering with the conformation and the orientation of the SP domain [34]. An experimentally determined structure showing the interface between FIXa and FVIIIa is needed to assess the role of the EGF2 domain in direct binding to FVIIIa.

### 3.4. Activation Peptide

The inactive zymogen FIX is converted to the enzymatically active FIXa after the release of the activation peptide [26]. This activation involves a double proteolytic cleavage to remove the activation peptide by FXIa or the TF/FVIIa complex [48]. The first cleavage between residues Arg191–Ala192 yields an intermediate of FIXa, and the second cleavage between Arg226–Val227 generates FIXa [9]. Because the structural information of the FIX zymogen is still lacking, the molecular details of AP in FIX activation remain elusive. There are several PTMs in the AP of FIX, including N-linked and O-linked glycosylation, sulphation, and phosphorylation [16,49,50,51]. The modifications of N-linked and O-linked glycosylation may contribute to the high solubility of FIX, extend its half-life, and suppress its premature cleavage and activation. The sulphation and phosphorylation show a correlation with protein recovery in vivo [16,51].

### 3.5. Serine Protease Domain

The catalytic domain of FIX is a typical member of the S1-type serine-protease family [26]. The SP domain is divided into two subdomains, composed of two independently folded β-barrels [52] (Figure 3B). Subdomain 1 (227–338 aa) contains an intra-domain disulfide bond (Cys252–Cys268), as well as an inter-domain disulfide bond (Cys178–Cys335) that covalently links the heavy and light chains after the activation peptide release (Figure 3A). Subdomain 2 (339–461 aa) contains two disulfide bonds (Cys382–Cys396 and Cys407–Cys435). Compared to subdomain 1, the β-barrel 2 of subdomain 2 is rotated by approximately 90° relative to β-barrel 1 (Figure 3B). The inter-subdomain contact is further clamped by the C-terminal helix (447–461 aa). Both subdomains contribute to the formation of the catalytic triad, His267, Asp315, and Ser411 [52] (Figure 3B).

Properly binding the FIXa’s SP domain to FVIIIa is required for FIXa to gain full enzymatic activity [26]. Several studies have shown that residues in the 339-helix (Lys339–Lys347), those in the 378-helix (Asp378–Arg384), and Asn392 of FIXa bind to FVIIIa [53,54,55,56] (Figure 3A). A model predicted that the A2-domain and A3-domain of FVIIIa is the key interaction site with 339-helix and 378-helix of the SP domain [34]. Furthermore, the SP domain of FIXa affords multiple binding sites, including an FX recognition site [38,39], a heparin-binding site [57], a high-affinity Ca^2+^ binding site [58], and a putative Na^+^-binding site [38,59].

### 3.6. Activation of FIX

FIX is a serine protease that circulates in an inactive zymogen form in the blood. In the blood coagulation cascade, the inactive zymogen FIX is activated to FIXa through a double proteolytic cleavage by FXIa in the intrinsic coagulation pathway or by the TF/FVIIa complex in the extrinsic pathway, releasing the activation peptide [60]. FIXa contains an N-terminal light chain and a C-terminal heavy chain. The heavy chain contains the SP domain and the light chain contains the Gla domain and two EGF domains. Proteolytic cleavage by FXIa or the TF/FVIIa complex does not fully account for FIX activation, as the activity of FIXa alone is extremely poor to cleave the physiological substrate FX [61]. To gain its full enzymatic activity, FIXa needs to form a Ca^2+^-dependent complex with the cofactor FVIIIa on phospholipid-containing membranes, known as the intrinsic Xase complex, which increases its activity by more than 200,000-fold [38,62].

## 4. Distribution of *F9* Mutations in Hemophilia B

Mutations in the *F9* gene have served as an outstanding model to the understanding of hemophilia B. According to an interactive database (http://www.factorix.org/, 20 January 2022), a total of 1094 unique mutations located at the *F9* gene locus have been reported in 3713 hemophilia B patients [41]. These mutations occur in coding and noncoding regions (including promoter, introns, and 3′ untranslated regions (UTR)) of the *F9* gene. Of the 1094 unique mutations, 897 (82%) mutations are found in the coding regions, 171 (15.6%) in noncoding regions, and 26 (2.4%) in multiple regions, and the corresponding patients retrieved in the database were 3218, 410, and 85, respectively (Table 1). Additionally, the mutations in the coding region were almost evenly distributed among the Gla, EGF1, EGF2, and SP domains, but occurred rarely in the activation peptide [41].

There are several types of *F**9* mutations, including point mutations, deletions, insertions, duplications, indels, neutral polymorphisms, and complex changes. Among these, point mutations (73.1%) are predominant in the *F9* gene locus, followed by deletion (16.5%) and other variant types (Table 1). Among the 3271 patients retrieved in the database, who carry 800 unique point mutations, 689 unique mutations occur in the *F9* coding sequence (Table 2). We analyzed the length of the coding sequence and number of patients with point mutation in each exon and found that the ratio of patient number/length of the coding sequence (PN/LCS) was higher in exon 2 and exon 8 than in the other exons, and that exon 1 had the lowest ratio (Table 3). In most diseases, pathogenic mutations were relatively concentrated in the conserved functional domains of a protein. The higher ratio of PN/LCS in exon 2 and exon 8 suggests that the corresponding encoding domains, propeptide, Gla domain, and SP domain, are more important for the function of FIX than the other domains.

## 5. Mechanisms of FIX Deficiency

*F9* mutations lead to FIX deficiency at multiple levels, including its gene structure, gene transcription, splicing, translation, posttranslational modifications, protein folding, and formation of functional complex [8,41,60,63,64,65,66,67]. In the *F9* variants database, about 88% of patients carry point mutations, and only about 12% have deletions, insertions, duplications, or indels (Table 1 and Table 2). Most deletions, insertions, duplications, and indels in the coding sequence cause a frame shift [41] that generates a truncated or an extended polypeptide with a changed sequence. A small portion of these mutations lead to the inframe effect, which has a deletion or (and) insertion with multiples of three nucleotides. Most patients with frame shift and inframe mutations show severe hemophilia B [41]. In the introns, deletions, indels, and insertions usually cause aberrant splicing, leading to severe hemophilia B in almost all of the affected patients [41]. About 2% of unique mutations affect multiple regions of the *F9* gene, and correspond to gross deletions of the *F9* gene, which also lead to severe hemophilia B. Additionally, it is worth noting that individuals with gross deletions have the highest risk (43%) of inhibitor development [9].

For hemophilia B patients with point mutations, their bleeding phenotypes vary from severe to mild, and there are multiple mechanisms causing FIX deficiency [8,60,66,67]. Generally, point mutations in the promoter region result in hemophilia B Leyden [9], those in the exons cause missense, nonsense, or silent mutations, and those in introns cause aberrant splicing [41]. We will focus on the pathogenic mechanisms of point mutations in the following sections.

### 5.1. Point Mutations in Noncoding Region

#### 5.1.1. Mutations in *F9* Promoter

Mutations in the promoter region of the *F9* gene often result in hemophilia B Leyden, which was first recognized in 1970 [63]. Individuals with hemophilia B Leyden have low FIX:C levels at birth, while their FIX:C levels rise rapidly at puberty, even to normal levels at adulthood [68]. More than 20 genetic mutations have been identified to be Leyden mutations [68]. These mutations distribute in a short region of the proximal promoter from c.-50 to c.-18, and cluster at three particular regions, including mutations in nucleotides c.-34 and c.-35, around nucleotide c.-49 and around nucleotide c.-19 [68]. Mutations in nucleotides c.-34 and c.-35 account for more than half of the cases of hemophilia B Leyden [9]. Depending on the mutations, some individuals initially exhibit severe hemophilia B, whereas others have a milder phenotype [41,69].

The mechanism of a progressive rise in the FIX:C level was originally thought to be related to the androgen receptors, which bind to the androgen-response element (ARE) binding site in the *F9* promoter to regulate gene transcription [70]. However, the ARE binding site is from c.-65 to c.-51 [41,70], which does not overlap with the site of Leyden mutations. Later, three transcription factor binding sites were identified in the promoter region with hemophilia B Leyden mutations, including hepatic nuclear factor 4α (HNF4α), CCAAT/enhancer-binding protein (C/EBP), and HNF6 [71,72]. A transgenic mice model of hemophilia B Leyden demonstrated that growth hormone (GH) was directly responsible for the puberty onset recovery of FIX production, which indicated a critical role of GH signal transduction in Leyden mutations [73]. It remains elusive how GH regulates the transcription factors.

Different from hemophilia B Leyden, the mutation at c.-55 in the promoter region was reported to cause hemophilia B Brandenberg, which has a stable and low FIX:C level throughout life [74]. It remains unclear the binding of which transcriptional factor is affected by the mutations. Additionally, the phenotype of several mutations at the *F9* promoter has not been established [41].

#### 5.1.2. Mutations in Introns

In the *F9* database, 226 patients had been reported with point mutations in the introns, accounting for 6.08% of the total patients (Table 2). Most of the intron point mutations were near the splice site (within 25 bp), and deep intron mutations were rarely reported in hemophilia B [9,41]. It has been reported that a mutation (c.392-569C>A) lies deeper in intron 4 of *F9* with a severe phenotype, however, the variant has not been validated as causing the disorder [9]. Point mutations in the intron usually lead to aberrant splicing, which causes deletion or insertion in the mRNA sequence and consequently affects the expression of the functional protein. According to the *F9* database, most hemophilia B patients with intron mutations show severe to moderate phenotypes [41], which is consistent with this mechanism. We noticed a mutation of c.520+13A>G in intron 5, with which most of the patients varied from moderate to mild phenotypes [41]. The differences suggest that this mutation may interfere with alternative splicing to various extents in different patients, and there should be a certain level of normal splicing.

#### 5.1.3. Mutations in 3′UTR

Twenty-two hemophilia B cases were reported with point mutations in 3′UTR [41]. The most reported mutation was c.2545A>G, with severe or moderate phenotypes. It had been proposed that this mutation leads to activation of a cryptic splice site that bypasses the polyadenylation signal, presumably destabilizing the mRNA or altering the splicing of the previous intron [75]. However, further experimental data are needed to support this proposition.

### 5.2. Point Mutations in Coding Region

Of the 1094 unique mutations in the *F9* database, 689 point mutations have been retrieved in the coding region. These mutations cover 336 residues of 461 residues in the FIX precursor, and can cause silent, nonsense, and missense mutations.

#### 5.2.1. Silent Mutations

Silent mutation is also called synonymous mutation, which alters a nucleotide but not the encoded amino acid. The majority of silent mutations in the *F9* gene appear neutral clinically, while some could influence protein production due to aberrant splicing, mRNA instability, or abnormal translation [76]. Sixteen unique silent mutations have been retrieved in 46 patients with severe to mild hemophilia B in the *F9* database [41], however, the underlying deleterious mechanisms have not been completely understood. Among these, several mutations near the end of exons have been proposed to cause a pathogenic effect at the stage of splicing control [9]. Exon-trap analysis showed that a case with c.87A>G (p.Thr29Thr) mutation resulted in nearly complete (99.1%) aberrant splicing (r.83–88del) [64]. The aberrant *F9* mRNA was translated into a mutant FIX (p.Cys28–Val30delinsPhe) with an in-frame mutation at the signal peptide cleavage site, which was abnormally retained in the ER and not secreted to the extracellular. A small amount (0.9%) of mutant *F9* r.87A>G translating into wildtype (WT) was also observed. Several cases with mutation c.459G>A (p.Val153Val) showed mild hemophilia B [77], and this mutation may impede translation, affect the protein conformation of FIX, and result in decreased extracellular protein level [78].

#### 5.2.2. Nonsense Mutations and Ribosome Readthrough

Nonsense mutations account for about 11% of point mutations, and usually cause severe hemophilia B. Individuals with nonsense mutations have increased risk of inhibitor development [9]. We noticed, however, a few patients with moderate or mild phenotypes [41], suggesting that spontaneous ribosome readthrough may occur with some nonsense mutations. Plasma from hemophilia B patients with the p.Arg294* and p.Arg298* mutations revealed traces of full-length FIX [79]. The in vitro expression also demonstrated a detectable level of secreted full-length FIX antigen for p.Arg162*, p.Arg294*, and p.Arg298* mutations [79]. A drug-induced readthrough model proposed for two mechanisms of the functional rescue: (1) prevalent reinsertion of the authentic residue (tryptophan), reverting the nonsense effects for the p.Trp240* and (2) gain-of-function for the p.Arg384* probably by readthrough-mediated missense variant (p.R384W) [80]. The amino acid substitutions in the ribosome readthrough depend on combinations of favorable FIX mRNA sequence and protein features. Ribosome readthrough in nonsense mutations that enable a low level of protein production has impacts on both disease severity and the likelihood of inhibitor development [80].

#### 5.2.3. Missense Mutations in Signal Peptide and Propeptide

The signal peptide and propeptide are regulatory sequences, which are cleaved off in mature FIX. Our recent study showed that missense mutations in the signal peptide caused FIX deficiency by interfering with FIX’s co-translational translocation to ER, or signal peptide cleavage [8]. Mutations in the hydrophobic region of the signal peptide, such as p.Ile17Asn, p.Leu20Ser, p.Leu23Pro, and p.Leu24Pro, disrupt co-translational translocation to ER (Figure 2, step ②), and thus lead to dysfunction of FIX secretion. On the other hand, mutations in the signal peptidase recognition site, such as p.Ala26Asp and p.Cys28Arg/Tyr/Trp, interfered with signal peptide cleavage (Figure 2, step ④), resulting in a longer propeptide that in turn affected FIX secretion (p.Ala26Asp), expression (p.Cys28Arg/Tyr/Trp), or γ-carboxylation of the Gla domain (Figure 2, step ⑤; p.Ala26Asp and p.Cys28Arg/Tyr/Trp).

The subdomain of propeptide can be sub-divided into two elements: the GGCX recognition site and the propeptidase recognition site [20]. A recent study showed that propeptide mutations caused FIX deficiency via several different mechanisms [8]. In the GGCX recognition site, the mutation p.Val30Ile decreased the protein expression, p.Ala37Asp results in un-carboxylated Gla domain (Figure 2, step ⑤), p.Ala37Thr mainly decreases the γ-carboxylation efficiency, and p.Ala37Val mainly affects protein secretion. Mutations in propeptidase recognition site, such as p.Arg43Gln, p.Arg43Trp, p.Arg46Ser, lead to an uncleaved propeptide in mature FIX (Figure 2, step ⑦) that interferes with the binding of Gla domain to phospholipid membrane [21,22,81]. With the use of a conformation specific antibody to detect the calcium-stabilized Gla domain, the uncleaved propeptide was found to disrupt the structure of Gla domain [8].

#### 5.2.4. Missense Mutations in Gla Domain

The Gla domain is involved in the FIXa binding to the phospholipids membrane, to TF in the TF/FVIIa complex, and to the C2 domain of FVIIIa [32,34], all of which are important to the full enzymatic activity of FIXa. In hemophilia B patients, naturally occurring point mutations are widely distributed in the Gla domain of FIX. Of the 12 glutamate residues, point mutations in nine residues have been reported in hemophilia B patients, and most of them showed severe bleeding phenotype [41]. Based on the structure of FIX Gla domain [25], these mutations should lead to destabilization of the ω-loop by disrupting its Ca^2+^ ion binding (Figure 3A). Most mutations in other residues at the ω-loop may also disturb the structure of this loop. For example, Asn48 is a structurally important residue for ω-loop, as its side chain forms hydrogen bonds with Gla54 and Gla73 and it is a ligand for a Ca^2+^ ion [25]. However, Lys51 is a surface-exposed residue and its side chain does not have a structural role in stabilizing the Gla domain. It has been proposed that this residue participates in binding the phosphate group of phosphatidylserine [24] and the collagen IV [35,82]. Additionally, Phe55, Phe71, and Arg75 in the Gla domain may be at the FVIIIa binding surface [25]. Thus, point mutations in these residues should affect the interaction between FIXa and FVIIIa. In summary, point mutations in the Gla domain disrupt the structural integrity of this region and affect the function of FIXa by weakening its interaction with the phospholipids membrane, TF, FVIIIa, or (and) collagen IV.

#### 5.2.5. Missense Mutations in EGF1 and EGF2 Domains

Disulfide bonds maintain the structural integrity of the EGF domains (Figure 3A). Missense mutations of the cysteines break the disulfide bonds and cause a severe bleeding phenotype [41]. These cysteine mutations in hemophilia B patients are associated with reduced FIX antigen level, suggesting that they destabilize the FIX [41]. Besides the disulfide bonds, Ca^2+^ ion binding to the EGF1 domain is essential for stabilizing its conformation (Figure 3A) and for the assembly of the Xase complex on the phospholipid surface. Asp93, Gln96, and Asp110 in the EGF1 domain are designated for Ca^2+^ ion binding [37]. Missense hemophilic mutations in these residues [41,83,84] may disturb the stability of EGF1 domain, which serves to correctly position the SP domain for optimal interaction with FVIIIa [44].

Generating the full enzymatic activity for FIXa requires its binding to FVIIIa. Several hemophilic mutations in EGF domains may interfere with FIXa binding to FVIIIa [34,66]. A model of the Xase complex suggested that the conserved residues, Ile112, Tyr115, and Trp118, in the EGF1 domain, and Ile136, Asn138, and Arg140 in the EGF2 domain contribute to FVIIIa binding [39]. However, mutations in the EGF2 domain mainly affect FIXa binding to FVIIIa by hindering the interaction of FIXa’s SP domain with FVIIIa [44]. Additionally, hemophilia B patients with missense mutations in residues Ile136 and Val153 of the EGF2 domain may disrupt the interaction of FIXa with the activated platelet surface and the assembly of the Xase complex on activated platelets.

In the extrinsic coagulation pathway, activation of FIX is triggered by its interaction with the TF/FVIIa complex. A study showed that FIX are barely activated in hemophilia B patients with p.Gly94Arg and p.Gly94Val mutations that disrupt the interaction of FIX with TF/FVIIa complex [85], indicating that the EGF1 domain is essential for this interaction. In a binding model of FIX/TF/FVIIa, Asp95, Glu98, Ser99, Asn104, Phe123, Asn127, and Glu129 on the surface of the EGF1 domain were proposed to be the key determinants for FIX binding to TF [32] (Figure 3A). Consistent with this model, numerous missense mutations in these residues have been reported to cause hemophilia B [41]. Experimental evidence, however, is required to establish the role of these residues in TF binding.

#### 5.2.6. Missense Mutations at Cleavage Site of Activation Peptide

More than 300 hemophilia B patients in the *F9* database have been reported with mutations at the cleavage site of activation peptide, namely at residues Arg191 or Arg226 [41,86]. However, hemophilia B patients are rarely reported to have mutations in the other residues of the activation peptide, and only seldom have polymorphisms been retrieved. Removal of the activation peptide requires cleavages at both the Arg191 and Arg226 site, and the missense mutations disrupt these cleavages [86]. Seminal studies showed that the cleavage only at Arg191 resulted in the loss of FIXa catalytic activity [26,87], whereas the cleavage only at Arg226 retains the catalytic activity but results in suboptimal binding to FVIIIa [48,86]. Consistently, most patients with missense mutations at Arg191 are associated with a moderate or mild bleeding tendency [41,88,89,90], whereas those at Arg226 have a severe bleeding phenotype [41,91,92,93]. In contrast, the p.Arg226Lys mutation that retains a positively charged residue does not appear to cause hemophilia B, as the variant does not hinder the release of the activation peptide [86].

Patients with mutations at Arg191 and Arg226 have different FIX antigen levels. Mutations at Arg226 residues are associated with normal or increased FIX levels, whereas Arg191 mutations are characterized by normally or modestly reduced antigen levels [86]. These observations show the detrimental effects of Arg191 mutations on the protein folding, secretion, or scavenging processes [6], all of which contribute to decreased antigen levels in the FIX. In turn, these features shape the bleeding phenotypes of patients, and produce the graded moderate to mild bleeding severity in hemophilia B patients with the Arg191Cys>Leu>Pro>His mutations [86].

#### 5.2.7. Missense Mutations in SP Domain

Among the hemophilia B patients with missense mutations, more than 55% have missense mutations located in the SP domain of FIX [9,41], emphasizing the importance of this domain. Cysteine residues involved in disulfide bond formation in the SP domain are important for stabilizing the protein folding (Figure 3), and missense mutations in these residues cause severe hemophilia B. The SP subdomain 1 contains a high affinity Ca^2+^ ion binding site (277–311 aa) at a surface loop [39] (Figure 3). Mutations in Arg294, Arg298, and Asn310 of the Ca^2+^ ion binding loop show significantly decreased FIX antigen levels [41,94,95,96], implying that the calcium loop affects the stability of the SP domain.

Impaired interaction with FVIIIa is another mechanism causing hemophilia B for missense mutations in the SP domain, especially those in subdomain 2 [41]. The 378-helix in SP subdomain 2 was thought to bind to FVIIIa [34] (Figure 3A). All known sequences from different species in this surface-exposed helix of FIXa are identical, and missense mutations in eight of the nine residues in this region cause hemophilia B [97], because mutations in this helix reduce FIXa’s affinity to FVIIIa. In addition, residues Lys339 in 339-helix and Asn392 were proposed to bind the A2 subunit of FVIIIa [39]. Subdomain-2 also contains an autolysis loop (356–369 aa), and mutations in this loop result in diminished binding of FIXa to FVIIIa. For example, mutation at Lys362 results in a substantial reduction in the rate of FX activation in the presence and absence of FVIIIa [98].

Additionally, the cleft between both SP subdomains forms the active site pocket that contain the catalytic triad of His267, Asp315, and Ser411 [39] (Figure 3B). Hemophilia B associated missense mutations in and around the active site pocket impairs the active site formation or substrate recognition [41].

## 6. Summary

Hemophilia B is one of the most heavily studied genetic disorders. With the wide availability of genetic diagnosis, more than 1000 unique mutations have been identified in patients, but the molecular mechanisms of FIX deficiency caused by these mutations have not been fully understood. In this review, we systematically summarize the structural and functional characteristics of FIX and the pathogenic mechanisms of the mutations that have been identified to date. Phenotypic heterogeneities are notable in patients with the same mutations, suggesting that the mutations are not the sole determinant of the bleeding phenotypes. Investigating the underlying pathogenic mechanisms will provide insights into such heterogeneity. For instance, our recent study showed that the vitamin K level may contribute to the heterogeneity of bleeding tendency in patients carrying mutations in signal peptide and propeptide regions [8]. Based on this, we proposed that the oral administration of vitamin K may alleviate the severity of their bleeding tendencies.

In our opinion, although current studies focus mainly on the clinical applications of the prophylaxis and gene therapy [3,99,100], more efforts should be devoted to uncovering the molecular mechanisms of FIX deficiency, which may lead to new strategies of precisely treating hemophilia B based on various mechanisms. For nonsense mutations, it is promising to develop new drugs to induce ribosome readthrough, which permits the secretion of full-length FIX [80]. Building on the encouraging results of treating spinal muscular atrophy by modifying aberrant splicing in the *SMN2* gene [101], it may also possible to develop small molecular drug for the hemophilic mutations that cause aberrant splicing. As for the missense mutations that affect the binding of FIXa with FVIIIa, a bi-specific antibody, similar to those used in hemophilia A [102], can be developed to treat hemophilia B. In brief, understanding the molecular mechanisms of FIX deficiency may improve future patient diagnosis and clinical decision making, and lead to precision medicine for hemophilia B patients.

## Figures and Tables

**Figure 1 ijms-23-02762-f001:**
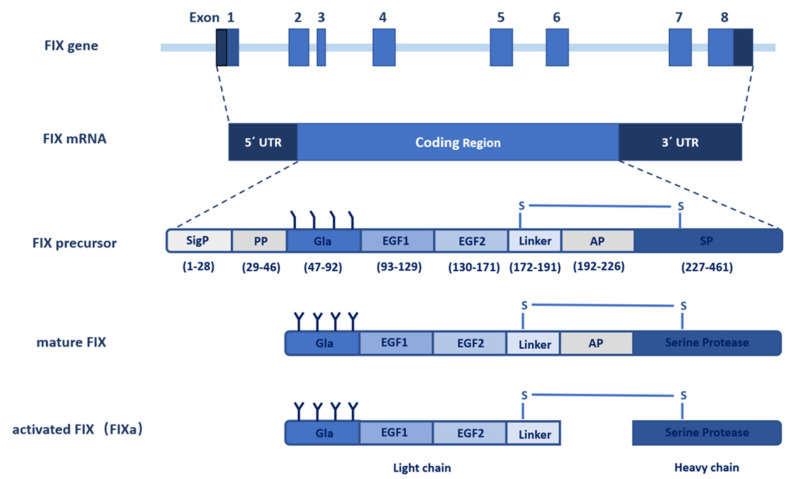
Schematic diagram of the *F**9* gene structure and FIX protein processing. The *F**9* gene contains eight exons and seven introns that transcribe into a 2.8 kb mRNA with 5′ and 3′ noncoding flanking sequence. The coding sequence translates into the FIX precursor, including the signal peptide, propeptide, Gla domain, EGF1 domain, EGF2 domain, linker, activation peptide (AP), and serine protease (SP) domain. The FIX precursor undergoes multiple post-translational modifications, especially γ-carboxylation, and is further processed to form the mature FIX secreted to the extracellular space. The inactive FIX zymogen is activated by the cleavage of the activation peptide to FIXa, which contains a light chain and a heavy chain linked by a disulfide bond.

**Figure 2 ijms-23-02762-f002:**
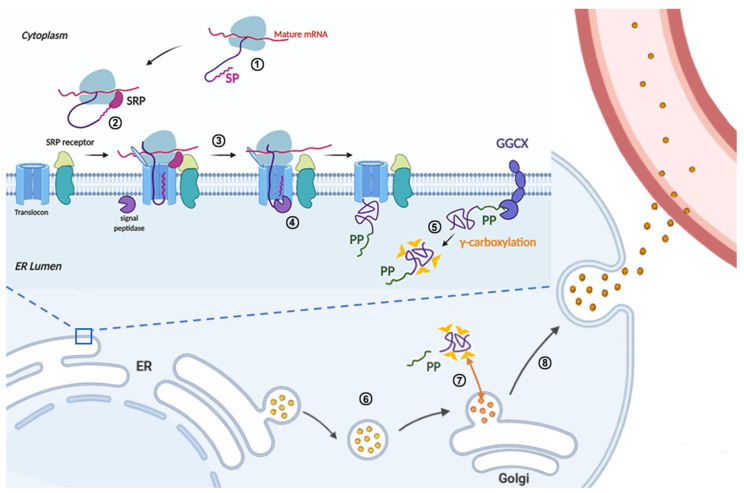
Schematic diagram of the FIX protein synthesis in liver cells. FIX protein synthesis is initiated in the cytoplasm (**①**), when the N-terminal signal peptide is synthesized, the signal recognition particle (SRP) recognizes the nascent signal peptide and stops the translation of FIX (**②**). Directed by SRP, the translation complex is translocated from the cytoplasm to the endoplasmic reticulum (ER) membrane (**②**), then the ribosome restarts the translation and the nascent polypeptide of FIX is co-translationally translocated into the ER lumen of liver cells (**③**). At the same time, the signal peptide is cleaved by signal peptidase in the ER lumen side (**④**). In the ER lumen, the propeptide of FIX directs the γ-carboxylation of glutamate residues in the Gla domain of FIX by an ER integral membrane protein, γ-glutamyl carboxylase (GGCX) (**⑤**). The fully γ-carboxylated FIX is transported to the Golgi apparatus by membrane coated vesicles (**⑥**), then the propeptide is cleaved by propeptidase (PACE/Furin) to generate mature FIX in the Golgi apparatus (**⑦**). The mature FIX is secreted into the blood (**⑧**). During FIX processing in the ER and Golgi apparatus, multiple post-translational modifications are added, such as N-linked glycosylation, O-linked glycosylation, and β-hydroxylation.

**Figure 3 ijms-23-02762-f003:**
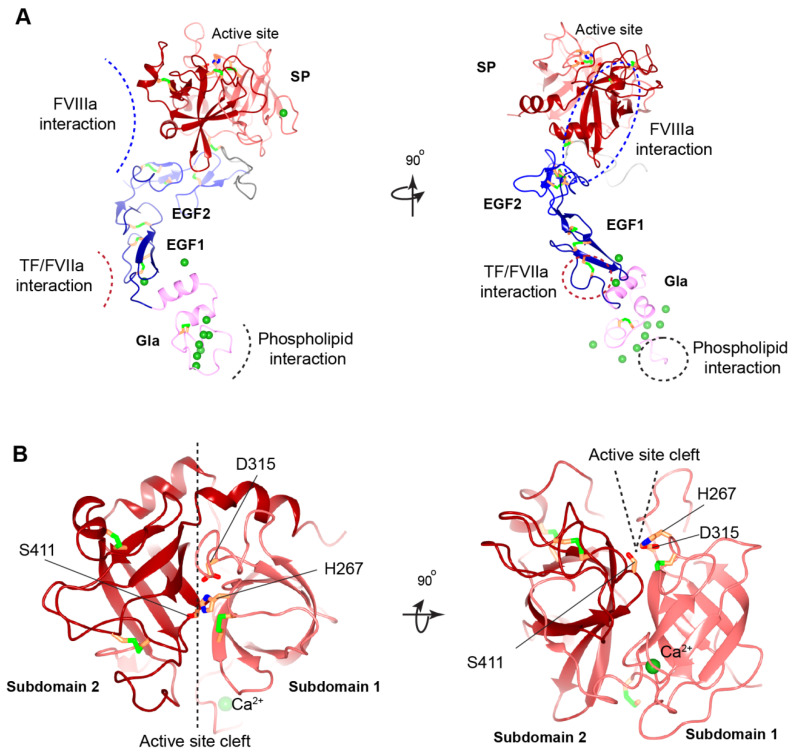
Structural model of human FIXa. (**A**) Overview of human FIXa’s structural model. The UniProt accession number of human FIX protein is P00740. The model is constructed in the following ways. The overall architecture is generated by Alphafold prediction [36]. Next, three partial human crystal structures, Gla domain (PDB: 1NL0) [25], EGF1 domain (PDB: 1EDM) [37], and combining EGF2-SP domain (PDB: 2WPH) [38] are superimposed onto the Alphafold model and a porcine FIXa crystal structure (PDB: 1PFX) [39] is superimposed to correct the inter-domain orientation between heavy and light chains. These superimposed crystal structures are combined to generate the final model. The Gla domain, EGF domains, SP domain, and linker sequence are colored in pink, blue, red, and grey, respectively. The Ca^2+^ ions are shown in the green sphere, and disulfide bonds in the green bars. The predicted interaction interfaces in FIXa with FVIIIa, with TF/FVIIa, and with membrane phospholipid are indicated by dashed curves (**left**) or dashed circles (**right**). The ω-loop in the Gla domain is also indicated by a dashed circle (**right**). (**B**) Structural model of human FIXa’s SP domain. Subdomain 1 and subdomain 2 are colored in light red and dark red, respectively. The active site cleft is formed between the two subdomains and side chains of the catalytic triad (His267, Asp315, and Ser415) are shown. These figures are generated with CCP 4 mg [40].

**Table 1 ijms-23-02762-t001:** Statistics of mutations in the *F9* database.

Regions		MTs	UMs	% of UMs	PN	% of PN
Noncoding	Promoter *	Point	23	2.10	86	2.32
		Deletion	2	0.18	2	0.05
		Polymorphism	5	0.46	5	0.13
	Intron	Point	86	7.86	226	6.09
		Deletion	14	1.28	23	0.62
		Insertion	2	0.18	8	0.22
		Indel	1	0.09	1	0.03
		Polymorphism	33	3.02	33	0.89
	3′ UTR	Point	2	0.18	22	0.59
		Duplication	1	0.09	1	0.03
		Polymorphism	2	0.18	3	0.08
Coding		Point	689	62.98	2937	79.10
		Deletion	145	13.25	204	5.49
		Insertion	33	3.02	38	1.02
		Indel	15	1.37	17	0.46
		Duplication	4	0.37	4	0.11
		Polymorphism	11	1.01	18	0.48
Multiple Regions		Deletion	19	1.75	78	2.10
		Insertion	1	0.09	1	0.03
		Indel	1	0.09	1	0.03
		Complex	5	0.46	5	0.13
Grand total			1094	100	3713	100

MTs, mutation types; UMs, unique mutations; PN, patients number. *: Promotor region in this table refers to the genomic sequences located upstream of translation initiation (ATG) in exon 1 of *F9* gene. Multiple regions indicate that mutation involves coding and noncoding regions.

**Table 2 ijms-23-02762-t002:** Statistics of point mutations in the *F9* gene.

Regions	Mutation Effects	Unique Mutations	Patient Number	% of Total Patients
Promoter	Leyden/NA	23	86	2.32
Exons	Missense	586	2422	65.23
	Nonsense	87	469	12.63
	Silent	16	46	1.24
Introns	Splice	86	226	6.09
3′ UTR	NA	2	22	0.59
Grand total		800	3271	88.1

% of total patients means percentage of total patients in the *F9* database.

**Table 3 ijms-23-02762-t003:** Distribution of patients with point mutations in the exons of the *F9* gene.

	Exon 1	Exon 2	Exon 3	Exon 4	Exon 5	Exon 6	Exon 7	Exon 8
Missense	43	351	22	205	164	365	177	1095
Nonsense	5	85	6	16	37	33	5	282
Silent	7	0	0	0	20	7	8	4
PN	55	436	28	221	221	405	190	1381
PN/LCS	0.63	2.66	1.12	1.94	1.71	2	1.65	2.52

PN, patient number; LCS, length of coding sequence; PN/LCS means the ratio of patient number to the length of coding sequence.

## Data Availability

Not applicable.

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
