# Peer review of "The Molecular Basis of FIX Deficiency in Hemophilia B"

_ijms, 2022, doi:10.3390/ijms23052762_

Round 1

Reviewer 1 Report

In this review, Shen and colleagues give a very comprehensive literature review on haemophilia B: the gene of FIX, FIX protein, distribution of F9 mutations in haemophilia B and mechanisms of FIX deficiency. The title focusses on the last part of the manuscript entitled ‘mechanism of FIX deficiency’. In my opinion this is more an overview of the types of mutations and the severity of haemophilia B linked to it and not always a description of the mechanisms. Often, they speculate on the mechanism or state that underlying mechanisms are not completely understood. Therefore, I would suggest changing the title of the manuscript, so it does not focus on the molecular mechanisms.

The review is of interest, however, the English in this manuscript is not of publication quality. The review is of interest, however, the English in this manuscript can be improved.

I only have a few comments/suggestions:

  1. The authors start with giving a clear overview of haemophilia B, the F9 gene and the FIX protein processing. Figure 2 is a schematic diagram of FIX protein synthesis in liver cells. It would help to insert numbers in every step that can be used in de legend of the figure as well as in the text (3, 3.1).
  2. It would help to add a figure visualising the binding of proteins and ions to the different parts of FIXa.
  3. It would be helpful to also visualise the part ‘mechanism of FIX deficiency’.

Author Response

We greatly appreciate the insightful comments from reviewer and their careful reading of the manuscript. Please see our point-by-point responses below and tracked changes in the revised manuscript.

1) In this review, Shen and colleagues give a very comprehensive literature review on haemophilia B: the gene of FIX, FIX protein, distribution of F9 mutations in haemophilia B and mechanisms of FIX deficiency. The title focusses on the last part of the manuscript entitled ‘mechanism of FIX deficiency’. In my opinion this is more an overview of the types of mutations and the severity of haemophilia B linked to it and not always a description of the mechanisms. Often, they speculate on the mechanism or state that underlying mechanisms are not completely understood. Therefore, I would suggest changing the title of the manuscript, so it does not focus on the molecular mechanisms.

Response:  Thanks for the nice suggestion. We change the title to be “The molecular basis of FIX deficiency in hemophilia B”

2) The review is of interest, however, the English in this manuscript is not of publication quality. The review is of interest, however, the English in this manuscript can be improved.

Response: The language was edited by native English speaker.

3)The authors start with giving a clear overview of haemophilia B, the F9 gene and the FIX protein processing. Figure 2 is a schematic diagram of FIX protein synthesis in liver cells. It would help to insert numbers in every step that can be used in de legend of the figure as well as in the text (3, 3.1).

Response: Thanks for the nice suggestion. We added the numbers in Figure 2, and described in figure legends and cited in the text.

4) It would help to add a figure visualising the binding of proteins and ions to the different parts of FIXa.

Response: Thanks for the suggestion. We constructed a structural model of human FIXa to help understanding the relationship between structure and functions of FIXa.

5) It would be helpful to also visualise the part ‘mechanism of FIX deficiency’.

Response: The structural model will be helpful to understand the mechanisms of FIX deficiency.

Reviewer 2 Report

1) ABSTRACT (last sentence): “Understanding molecular mechanisms of FIX deficiency will provide significant insights for future patient diagnosis and treatment”.

The authors must explain how they think that the understanding of molecular mechanisms of FIX deficiency will provide significant insights for the diagnosis and treatment of these patients in the future.

2) In point 6 (SUMMARY) the authors state “Although hemophilia B is one of heavily studied genetic disorders, researches now mainly focusing on the clinical applications of the prophylaxis and gene therapy”.

The following paragraph (and reference) must be included just after the aforementioned sentence:

Outcomes of gene therapy in hemophilia have so far been encouraging in terms of levels and times of expression utilizing mainly adeno-associated vectors. Nonetheless, these therapies are associated with immunogenicity and hepatotoxicity. Optimizing the vector serotypes and the transgene (variants) will boost clotting efficacy, thus augmenting the viability of these protocols. In the case of hemophilia B, the priority should be to optimize both the vector serotype, diminishing its immunogenicity and hepatotoxicity, and the transgene, boosting its clotting efficacy so as to minimize the amount of vector administered and reduce the prevalence of complications without compromising the efficacy of the protein expressed.

Rodríguez-Merchán, E.C.; De Pablo-Moreno, J.A.; Liras, A. Gene Therapy in Hemophilia: Recent Advances. Int. J. Mol. Sci. 2021, 22, 7647. https://doi.org/10.3390/ijms22147647

Author Response

We greatly appreciate the insightful comments from reviewer and their careful reading of the manuscript. Please see our point-by-point responses below and tracked changes in the revised manuscript.

1) ABSTRACT (last sentence): “Understanding molecular mechanisms of FIX deficiency will provide significant insights for future patient diagnosis and treatment”. The authors must explain how they think that the understanding of molecular mechanisms of FIX deficiency will provide significant insights for the diagnosis and treatment of these patients in the future.

Response:  In the summary section, we added a paragraph to explain it simply. It will be easy to understand this sentence.

2) In point 6 (SUMMARY) the authors state “Although hemophilia B is one of heavily studied genetic disorders, researches now mainly focusing on the clinical applications of the prophylaxis and gene therapy”. The following paragraph (and reference) must be included just after the aforementioned sentence: Outcomes of gene therapy in hemophilia have so far been encouraging in terms of levels and times of expression utilizing mainly adeno-associated vectors. Nonetheless, these therapies are associated with immunogenicity and hepatotoxicity. Optimizing the vector serotypes and the transgene (variants) will boost clotting efficacy, thus augmenting the viability of these protocols. In the case of hemophilia B, the priority should be to optimize both the vector serotype, diminishing its immunogenicity and hepatotoxicity, and the transgene, boosting its clotting efficacy so as to minimize the amount of vector administered and reduce the prevalence of complications without compromising the efficacy of the protein expressed.

Response:We added several references for gene therapy, but did not add the paragraph as reviewer suggested. Because this review mainly focuses on mechanisms of FIX deficiency, but not gene therapy in hemophilia B. In addition, there are lots of reviews discussing about gene therapy, so here we did not need to talk a lot about the issue of gene therapy.